# Wing Geometric Morphometrics of Workers and Drones and Single Nucleotide Polymorphisms Provide Similar Genetic Structure in the Iberian Honey Bee (*Apis mellifera iberiensis*)

**DOI:** 10.3390/insects11020089

**Published:** 2020-01-30

**Authors:** Dora Henriques, Julio Chávez-Galarza, Juliana S. G. Teixeira, Helena Ferreira, Cátia J. Neves, Tiago M. Francoy, M. Alice Pinto

**Affiliations:** 1Centro de Investigação de Montanha (CIMO), Instituto Politécnico de Bragança, Campus de Sta. Apolónia, 5300-253 Bragança, Portugal; dorasmh@ipb.pt (D.H.); jchavez@undc.edu.pe (J.C.-G.); helenamf93@gmail.com (H.F.);; 2Escola de Agronomia, Universidad Nacional de Cañete, Urb. Rosa de Hualcará, Calle Canal Maria Angola s/n, San Vicente de Cañete, Lima 15701, Peru; 3Faculdade de Filosofia, Ciências e Letras de Ribeirão Preto, Universidade de São Paulo, Av Bandeirantes, 3900, Ribeirão Preto 14040-900, Brazil; s.julianat@gmail.com; 4Escola de Artes, Ciências e Humanidades, Universidade de São Paulo, Rua Arlindo Béttio, 1000, São Paulo 03828-000, Brazil; tfrancoy@usp.br

**Keywords:** Iberian honey bee, spatial population structure, spatial principal component analysis (sPCA), SNPs

## Abstract

Wing geometric morphometrics has been applied to honey bees (*Apis mellifera*) in identification of evolutionary lineages or subspecies and, to a lesser extent, in assessing genetic structure within subspecies. Due to bias in the production of sterile females (workers) in a colony, most studies have used workers leaving the males (drones) as a neglected group. However, considering their importance as reproductive individuals, the use of drones should be incorporated in these analyses in order to better understand diversity patterns and underlying evolutionary processes. Here, we assessed the usefulness of drone wings, as well as the power of wing geometric morphometrics, in capturing the signature of complex evolutionary processes by examining wing shape data, integrated with geographical information, from 711 colonies sampled across the entire distributional range of *Apis mellifera iberiensis* in Iberia. We compared the genetic patterns reconstructed from spatially-explicit shape variation extracted from wings of both sexes with that previously reported using 383 genome-wide SNPs (single nucleotide polymorphisms). Our results indicate that the spatial structure retrieved from wings of drones and workers was similar (r = 0.93) and congruent with that inferred from SNPs (r = 0.90 for drones; r = 0.87 for workers), corroborating the clinal pattern that has been described for *A. m. iberiensis* using other genetic markers. In addition to showing that drone wings carry valuable genetic information, this study highlights the capability of wing geometric morphometrics in capturing complex genetic patterns, offering a reliable and low-cost alternative for preliminary estimation of population structure.

## 1. Introduction

Morphological markers have long been used to understand the diversity of life and the relationships among different taxa [1]. However, with DNA-based technologies being increasingly affordable and accessible to laboratories with low budgets, molecular markers are overriding morphological markers to become the standard tool for interrogating organisms [2,3,4,5]. Compared with morphological markers, molecular markers provide higher taxonomic resolution, allow a deeper understanding of the evolutionary processes, are stable and detectable independently of growth, differentiation and development and are not confounded by the environment [4,5,6]. On the other hand, morphological markers are still less costly, easier to use, do not require sophisticated instruments and a high level of technical expertise [7,8]. Given these advantages, when morphological traits offer taxonomic resolution that goes beyond the species level allowing detection of fine genetic structure, they could be used to identify populations that can be later studied with molecular markers [7,8].

Traditional morphological methods typically employ multivariate statistics to quantitative variables such as length, height and width of structures, distances between certain landmarks and sometimes angles and ratios [1,9,10]. However, these methods are not always efficient in describing all the features of the body structures and the information of shape and size can be difficult to disentangle [9,10,11]. These shortcomings can be surpassed by geometric morphometrics, an approach developed at the end of the 20th century [1,9,10]. Geometric morphometrics uses 2D or 3D coordinates of biologically definable points, called landmarks, that capture the geometry of the structure under scrutiny [1,10]. To disentangle shape and size information and to remove variation in translation and rotation of the targeted structures, the landmarks are superimposed in a common coordinate system [1,9,10]. Compared with traditional methods, geometric morphometrics is more flexible in data acquisition, is able to capture the geometry of morphological structures preserving this information throughout the analysis [1] and has a greater discriminating power (see [12,13,14,15,16]). These features have made geometric morphometrics a popular tool to investigating diversity patterns of many organisms (reviewed in Adams, et al. [1] and Tatsuta, et al. [11]), the western honey bee (*Apis mellifera* L.) is no exception [2,16,17,18,19,20,21,22].

The father of *A. mellifera* taxonomy and biogeography, Friedrich Ruttner [23], applied traditional morphometrics to 36 body size, pilosity, colouration, and wing venation traits to distinguish 24 subspecies collected from Africa, Europe, and the Middle East. Since the pioneer work of Ruttner [23], a wide array of molecular and morphological (including wing geometric morphometrics) markers has been used to identify and classify honey bee lineages and subspecies in both the native and introduced ranges [2,3,17,18,21,22,24,25,26,27,28,29,30,31]. The patterns obtained with the more recently developed wing geometric morphometrics approach have been largely consistent with those inferred from the traditional morphometry [2,16,22]. However, this has not always been the case. As an example, while Ruttner [23] placed the North African *Apis mellifera intermissa* and *Apis mellifera sahariensis* together with the European subspecies *Apis mellifera mellifera* and *Apis mellifera iberiensis* in the western European (M) lineage, later phylogeographic studies using nuclear DNA markers [22,32,33,34] and geometric morphometrics [22] re-arranged the African (A) lineage to include North African together with sub-Saharan subspecies.

Wing geometric morphometrics has proved capable of identifying lineages, subspecies, and even hybrids [2,16,17,18,19,20,21,22]. However, whether this method is equally capable of capturing genetic structure within a subspecies range has yet to be fully assessed. The honey bee subspecies that evolved in the Iberian Peninsula, *A. m. iberiensis*, offers a powerful model system to address this issue for two main reasons. First, the Iberian honey bee exhibits a complex phylogeographic pattern which, contrarily to other subspecies, has been largely resilient to confounding beekeeper-mediated processes [35]. Second, no other subspecies has been as thoroughly surveyed as the Iberian honey bee. In the last 40 years, thousands of colonies sampled from across the entire Iberia have been interrogated using a battery of markers, including morphology [22,25,36], allozymes [25,37], mitochondrial DNA [25,32,38,39,40,41,42,43,44,45,46,47,48], microsatellites [22,32,39,49,50,51], SNPs [35,52], and even whole genomes [53]. Early surveys using traditional morphometrics and allozymes revealed existence of a smooth gradient extending from France to North Africa, with Iberian populations showing intermediate phenotypes [25,37]. However, this pattern was not supported by maternally-inherited mitochondrial polymorphisms which formed not a smooth but a steep northeast–southwest cline of highly divergent haplotypes of western European and African ancestry in Iberia [25,32,35,38,39,40,41]. Adding to the complexity, microsatellites [22,32,49,51] and wing geometric morphometrics data [22] did not capture either the Iberian northeast–southwest cline or the smooth gradient extending from north Africa to France, but instead a sharp disruption in genetic variation between the two continents. The lack of genetic structure reported from wing shape [22] and microsatellite [22,32,49,51] data contrasts with genome-wide SNPs [35] and even whole genomes [53], which recovered the clinal pattern reported earlier. 

The vast amount of knowledge generated by all those works enables unprecedented comparisons between markers as to their power in retrieving complex diversity patterns. Using the Iberian honey bee as a model system, the first goal of this study was to assess the efficiency of geometric morphometrics in capturing genetic structure within a subspecies range. To that end, a comprehensive geometric morphometrics dataset was integrated with geographical information to disentangle global structure from local structure and random noise. If geometric morphometrics would prove efficient in describing structure in a subspecies with a complex phylogeographic pattern, it could be used as a low-cost alternative for preliminary genetic surveys.

Geometric morphometrics can be applied to different parts of an organism, but wings, mainly from females, have been the major target in Diptera and Hymenoptera [11]. Insect wings are well suited to geometric morphometrics analysis because of the high heritability of shape [11,54] and of the approximate two-dimensional structure, which reduces digitizing errors [55]. A number of studies have applied geometric morphometrics to investigate wing sexual dimorphism in social hymenopterans [56,57,58], which tends to be pronounced in this group of insects [59] due to the haplodiploid sex-determination system. In honey bees, geometric morphometrics surveys have typically been applied to the forewings of workers [2,16,17,18,19,20,21,22] since drones usually exhibit a higher number of wing venation anomalies due to their haploid condition [60]. Besides, drones are only present in the colony during certain periods of the year, making harder their sampling. Whether the forewings of drones carry information similar to that of forewings of workers and whether this information is able to retrieve complex phylogeographic patterns is unclear. Hence, the second goal of this study was to assess the degree of similarity between structures inferred from the forewings of both sexes.

To achieve these goals, we used a geometrics morphometrics approach to examine the forewing shapes of multiple drones and workers belonging to 711 georeferenced colonies of *A. m. iberiensis* sampled across three north-south Iberian transects. The 711 colonies were previously analysed with 383 genome-wide SNPs genotyped in drones, which showed the existence of a strong global structure formed by two genetic clusters [35]. Using the spatially-explicit wing shape data obtained for drones and workers and the SNP data genotyped by Chávez-Galarza, et al. [35], we addressed the following questions: (i) Is the wing geometric morphometrics approach able to detect genetic structure in the Iberian honey bee? (ii) Are the results obtained with wing geometric morphometrics concordant with those inferred from SNP data? (iii) Are the patterns inferred from the forewings of drones and workers congruent? By answering these questions, we expect to provide new insights into the usefulness of drone wings and the power of geometric morphometrics in detecting genetic structure in honey bee populations.

## 2. Materials and Methods

### 2.1. Sampling

Samples of drones and workers were collected in 2010 from 711 colonies distributed in the Iberian Peninsula along three transects (Figure 1): the Atlantic transect (AT; 8 sites), the Central transect (CT; 9 sites), and the Mediterranean transect (MT; 6 sites). Each of the 23 sites was represented by 10 georeferenced apiaries. Drones and workers were collected from the inner frames of three hives per apiary and stored in absolute ethanol until subsequent analyses.

### 2.2. Geometric Morphometrics Analysis

For most of the 711 colonies, the right forewing was detached from five drones and five workers, placed between slides and photographed with a digital camera attached to a stereomicroscope. Specifically, for 56 colonies (37 drones and 19 workers) the number of detached forewings was lower than five (see Appendix A for details). In order to capture the Cartesian coordinates of the landmarks, a tps file was constructed from the images using tpsUtil v1.76 [61]. The tps file was used by tpsDig v2.17 [62] where 19 homologous landmarks were manually plotted across the forewing venation, following the order displayed in Figure 2. Wings missing one or more landmarks were removed from the dataset leading to a total of 3262 drone (709 colonies) and 3511 worker (710 colonies) forewings for downstream analyses (Appendix A). The landmark coordinates obtained from tpsDig were used as input in MorphoJ v1.06a [63]. The Cartesian coordinates of the specimens were then aligned to remove variations in size, position, and orientation using the generalized Procrustes superimposition method [64]. The mean values of the landmark coordinates obtained from multiple wings per colony (5 wings/per colony for 71.7% of drones and 95.6% workers; Appendix A) were used to generate a covariance matrix for downstream analyses. Pairwise Procrustes distances among populations (sites) were obtained through the canonical variate analysis (CVA). Pearson correlation coefficient (r) values were computed in R 3.6.0 [65] using as input data the Procrustes distances, obtained from drones and workers’ wing shapes (Appendix A), and the F_ST_ values (Appendix A) obtained from SNPs genotyped in drones by Chávez-Galarza, et al. [35].

### 2.3. Estimation of Spatial Structure

Spatial structure was inferred from drone and worker forewing shapes using the model-free multivariate spatial principal component analysis (sPCA). The sPCA of the drone SNPs was performed in the software package adegenet [66], using the K-nearest neighbours to model the spatial connectivity among individuals (see Chávez-Galarza, et al. [35] for further details). For wing geometric morphometrics data, the sPCA was carried out following the same parameters as for SNP data [35], but using the ade4 software package [67]. The morphometric input data used in the sPCA was constructed from the average distances calculated between proximal landmarks for each colony.

The sPCA is a modification of PCA and considers the genetic variance of individuals or populations together with their spatial autocorrelation measured by Moran’s I, which ranges from −1 to 1 [68]. To calculate Moran’s I, the genetic variance observed at a given location is compared with those at neighbouring sites [69]. Highly positive Moran’s I values indicate the presence of global structure, which occurs when each sampling location is genetically closer to neighbours than randomly chosen locations, as it happens in the presence of patches, clines or intermediate states. On the other hand, highly negative Moran’s I values indicate the existence of a stronger genetic differentiation among neighbours than randomly-chosen locations, which indicates local structure. The Moran’s I values were calculated considering the component with the highest eigenvalue. To evaluate the consistency of the detected geographical structures and the statistical significance for global and local structures, Monte Carlo simulations were implemented using 10,000 permutations, meaning that the significance threshold level is set at 1 × 10^−4^. This test validates whether genetic variance is distributed at random on the connection network (null hypothesis), or display global or local spatial structure (alternative hypothesis). In each permutation, the maximum of the mean of determination coefficient (max(t)) is computed. The coefficient of determination is estimated by linear regression between the matrix of allelic frequencies or morphometric measures and the eigen analysis of spatial connectivity among individuals. If the observed max(t) associated with the local or global structure is different from most simulated values and the *P*-value ≤ 1 × 10^−4^, the spatial distribution of the genetic variance is not random and the null hypothesis can be rejected. Spatial structure inferred from drone SNPs and wing geometric morphometrics data of both drones and workers were compared through the Pearson correlation coefficient (r) calculated in R 3.6.0 [65] using the first component of the sPCA.

## 3. Results

The canonical variate analysis (CVA) of drone and worker forewing shapes showed no clear structure in the Iberian honey bee (Figure 3). Pairwise Procrustes distances, which indicate the degree of differentiation among populations (sites) in wing shapes, were low and similar in the two sexes (Appendix A). Procrustes distance values ranged from 0.0037 to 0.0116, in drones, and 0.0030 to 0.0112, in workers. The most divergent populations originated from the latitudinal extremes of the Iberian honey bee range, namely: MT2 and AT6, for drones (Procrustes distance = 0.0116), and CT1 and AT7, for workers (Procrustes distance = 0.0112; Appendix A). Pairwise Procrustes distances of drones were moderate but significantly correlated with those of workers (r = 0.55, *P*-value < 2.2 × 10^−16^). Similar correlations were obtained between F_ST_ values inferred from SNPs (Appendix A) and Procrustes distances (Appendix A) inferred from wing shapes of drones (r = 0.50, *P*-value < 2.2 × 10^−16^) and workers (r = 0.58, *P*-value < 2.2 × 10^−16^).

The patterns of forewing shape variation of both sexes were further examined using a spatially explicit approach. The first global score (the axis with most positive eigenvalue) for drones and workers was associated with moderate autocorrelations (Moran’s I = 0.186 and Moran’s I = 0.304, respectively). On the other hand, the first local score (the axis most negative eigenvalue) was low either for drones (Moran’s I = −0.0178) or workers (Moran’s I = −0.0208). The interpolation of the first global score for both sexes detected a well-defined cline dividing Iberia along a northeast–southwest axis (Figure 4a,b), being the southwestern individuals (large black squares) the most differentiated from the northeastern individuals (large white squares). The global structure retrieved from wing geometric morphometric data was concordant with that obtained from SNPs genotyped in drones by Chávez-Galarza, et al. [35] (Figure 4c).

As reported in Chávez-Galarza, et al. [35] for SNPs, Monte Carlo simulations using wing shape data confirmed the existence of global structure in both drones (max(t) = 0.01627, *P*-value < 1 × 10^−4^; Figure 5a) and workers (max(t) = 0.0303, *P*-value < 1 × 10^−4^; Figure 5b). In contrast, there was no statistical evidence of local structure for either drones (max(t) = 0.00359, *P*-value = 0.8670; Figure 5c) or workers (max(t) = 0.00435, *P*-value = 0.2772; Figure 5d).

Spatial patterns inferred from wing shape data of drones and workers and from SNPs (reported in Chávez-Galarza, et al. [35]) were compared through the correlation coefficient (r) using the first component of the sPCA (Figure 4). Interestingly, high correlation values were obtained not only between wing shapes of both sexes (r = 0.93, *P*-value < 2.2 × 10^−16^) but also between SNPs and wing shapes of drones (r = 0.90; *P*-value < 2.2 × 10^−16^) and workers (r = 0.87; *P*-value < 2.2 × 10^−16^), suggesting that the spatial analysis is more powerful in detecting population structure as compared to the traditional CVA and Procrustes distances.

## 4. Discussion

The landmark-based geometric morphometrics is acknowledge as the most rigorous and powerful morphometric technique currently available. It has been applied to a variety of structures (e.g., wings, shells, cranial bones, mandibles, premaxilla) and organisms (from insects to mammals) to describe inter- and intra-specific variation and underlying evolutionary processes [70,71,72,73,74,75,76,77,78]. In *A. mellifera*, geometric morphometrics of wing vein junctions has been used to discriminate amongst evolutionary lineages and subspecies [2,16,18,19,21], but not to detect genetic structure within subspecies. Herein, we elected *A. m. iberiensis* as the model system to unequivocally show, for the first time in *A. mellifera*, the power of wing geometric morphometrics in retrieving complex phylogeographic patterns, particularly when wing shape data is analysed using a spatially explicit framework. 

We first analysed the wing shape data obtained from drones and workers using traditional multivariate statistics. This approach was unable to reveal a clear structure within the Iberian honey bee range (Figure 3), congruent with earlier surveys of microsatellites [22,32,39,49,50,51] and wing geometric morphometrics [22]. However, when the wing shape data were analysed using a spatially-explicit framework, we found statistical support for the presence of global structure (*P*-value < 1 × 10^−4^). Interpolation of the first global scores, computed from variations carried by the wing landmarks of both sexes, was able to unambiguously retrieve the clinal shape of variation (Figure 4) with a striking degree of overlap with that inferred from mtDNA and genome-wide SNPs [25,32,35,38,39,40,41,42,43,44,45,46,47,48,52]. The lack of genetic structure reported in previous works [22,51] has been explained by recent beekeepers-mediated gene flow associated with mobile beekeeping, erasing any signal of clinal variation [51]. However, these works missed the geographical breadth and the spatially-based statistical power needed to detect the cline reported here and in other studies using molecular markers [25,32,35,38,39,40,41,42,43,44,45,46,47,48,52]. The integration of geographical information with geometric morphometrics methods has also been used with success in other insects [71,72], further supporting the power of the spatial approach to uncover population structure.

Arias, et al. [25] nearly approached the Iberian cline using traditional morphometry on numerous traits measured in workers. However, wing geometric morphometrics has several practical advantages over traditional morphometrics. While Arias, et al. [25] needed to examine 23 characters from different parts of the honey bee body, wing geometric morphometrics only requires detachment of forewings. Due to their near two-dimensional structure, wings can then be easily mounted and photographed, or even digitalized, and stored forever for future analyses. Analysis of wing images is easier and less time consuming than measuring body size, pilosity, colouration, and lengths and angles of wing veins. The most labour-intensive step in geometric morphometrics is to manually plot the 19 landmarks in wings, but this task is expected to be fully automated in the near future.

This study revealed a high consistency between wing morphology and genome-wide SNPs, with correlation values of 0.87 (workers’ wing shape versus SNPs) and 0.90 (drones’ wing shape versus SNPs) for the spatial genetic patterns inferred from both markers. The ability of wing traits to capture a genetic pattern so strikingly similar to that inferred from hundreds of molecular loci spread across the 16 honey bee chromosomes [35,52] could be explained by a high heritability and a polygenic nature of wing shape. While the genetic basis of forewing venation is virtually unknown in most insects, including honey bees, it is likely that wing shape has a strong genetic control, as suggested by this and other studies which have also reported concordance between wing morphology and molecular markers, such as mtDNA and microsatellites [78,79,80,81,82,83].

In Hymenoptera, wing geometric morphometrics has been used to examine sexual dimorphism [56,57,58]. While Pretorius [57] found a good correlation (r = 0.47) between wing shapes of males and females in several species of Tachyspex, Abbasi [56] reported a weak correlation (r = 0.16) in three species of Polistes. Here, we showed, for the first time in *A. mellifera*, that the genetic information carried by drone wings for identifying genetic structure is similar to that of workers, the commonly used form in diversity studies [2,16,17,18,19,20,21,22]. Furthermore, the spatial structure inferred from SNPs was more similar to that inferred from drone wings (r = 0.90) than to that of workers (r = 0.87). This finding makes sense since the 383 genome-wide SNPs were genotyped in drones [35]. Despite the potential usefulness of drone wings for carrying out diversity studies, workers have been preferred because they are always available in a colony and they typically display a lower number of wing venation anomalies. In this study, a total of 206 drones and 14 workers were eliminated from the initial dataset because of missing landmarks. Drones are haploid, allowing more frequent phenotypic expression of deleterious alleles, which were identified here as missing landmarks. 

In this study, we examined concordance between genetic patterns captured by different genetic markers and sexes in a honey bees subspecies that is free of introgression from commercial strains [35,38,43,45,46,49,52,53] and that has been shaped by complex evolutionary processes [35,52,53], involving recurrent cycles of contraction, expansion, admixture, and adaptation, typical of long-term glacial refugia. Our findings suggest that geometric morphometrics of worker as well as drone forewings can be used reliably for detecting intra-subspecific genetic structure. The question whether the observed relationships are unique for this particular honey bee subspecies or represent a more general pattern deserves to be further investigated.

## 5. Conclusions

This study highlights the power of wing geometric morphometrics in capturing genetic structure within honey bee subspecies, especially when shape data is integrated with geographical information and analysed using spatial statistics. Spatial analysis of wing shape variation was able to detect a well-defined cline that bisects Iberia along a northeast–southwest axis, a pattern that remarkably parallels that of genome-wide SNPs. Moreover, in spite of the haploid nature of drones, allowing more frequent phenotypic expression of wing abnormalities, the genetic structure retrieved from drones was similar to that of workers, suggesting that drone forewings carry usable genetic information for diversity studies. Geometric morphometrics has the benefit over molecular methods of being cheaper and easier to use [7,18], offering a preliminary or complementary tool for identifying honey bee lineages, subspecies, and even detecting intra-subspecies structure [2,16,17,18,19,20,21,22].

## Figures and Tables

**Figure 1 insects-11-00089-f001:**
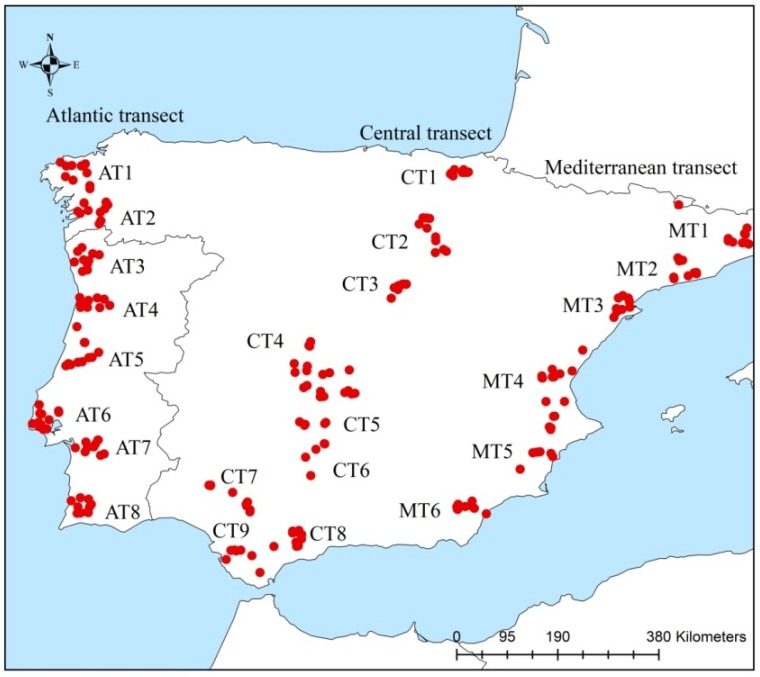
Geographical location of 711 colonies sampled in the Iberian Peninsula.

**Figure 2 insects-11-00089-f002:**
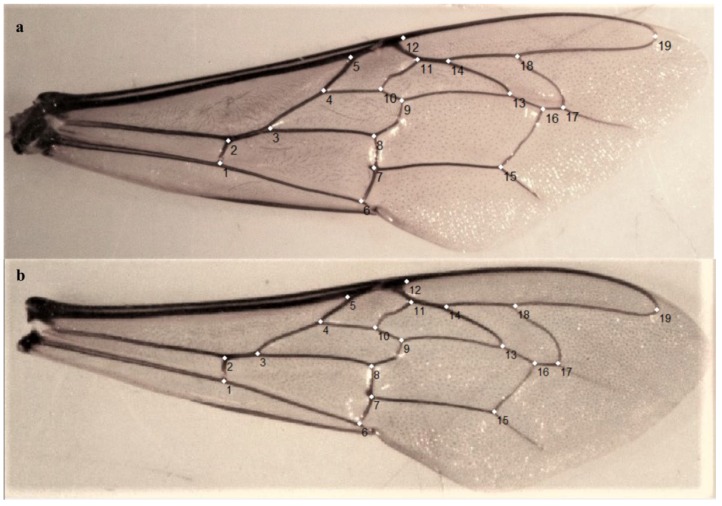
The 19 landmarks placed on the vein junctions of the right (**a**) drone and (**b**) worker forewings.

**Figure 3 insects-11-00089-f003:**
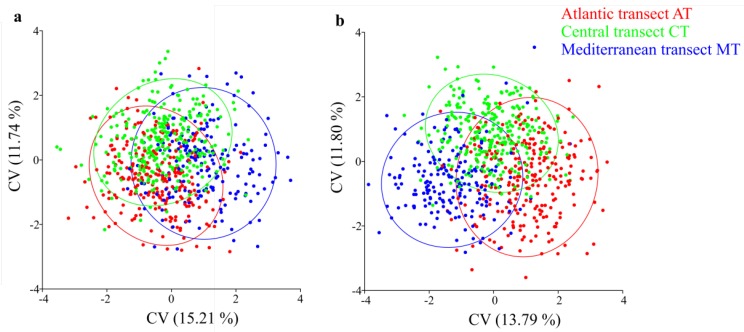
Scatterplots of individual scores from the canonical variant (CVA) analysis of (**a**) drone and (**b**) worker wing landmarks of the Iberian honey bee. Each dot represents a colony.

**Figure 4 insects-11-00089-f004:**
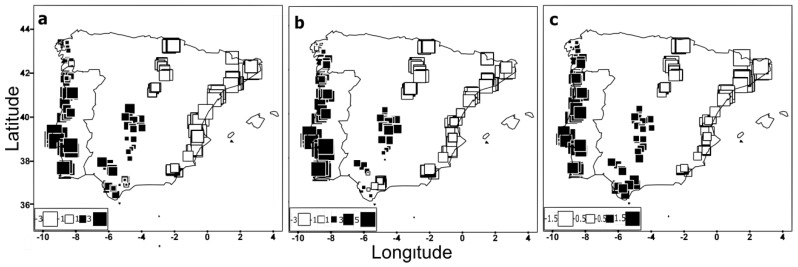
Global structure displayed by the 711 colonies (from 23 sampling sites located in the Atlantic, Central, and Mediterranean transects) after the spatial principal component analysis (sPCA). Global scores (first principal component) obtained from (**a**) wing geometric morphometrics of drones, (**b**) wing geometric morphometrics of workers, and (**c**) SNPs genotyped in drones by Chávez-Galarza, et al. [35]. Squares represent population scores and are spatially arranged according to the geographical coordinates of the colonies. Large black squares indicate colonies well differentiated from those denoted by large white squares whereas small squares indicate a lower degree of differentiation.

**Figure 5 insects-11-00089-f005:**
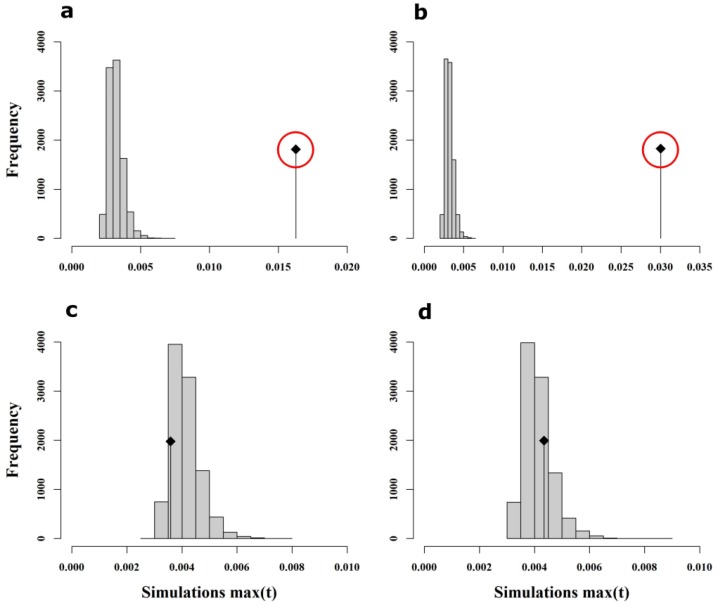
Results of Monte Carlo simulations using wing shape data. The x-axis represents max(t) calculated in each permutation whereas the y-axis represents the frequency of each max(t) class. The observed max(t) value is represented by the vertical black line with the black square on top. Monte Carlo simulations support the existence of global structure for both sexes (a-drones; b-workers) but not local structure (c-drones; d-workers), as indicated by the location of the observed max(t) value outside (**a**,**b**) or inside (**c**,**d**) the histogram of simulated values. Outside locations of observed max(t) values specify statistical significance (indicated by the red circle).

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
