# Peer review of "Wing Geometric Morphometrics of Workers and Drones and Single Nucleotide Polymorphisms Provide Similar Genetic Structure in the Iberian Honey Bee (Apis mellifera iberiensis)"

_insects, 2020, doi:10.3390/insects11020089_

Round 1
Reviewer 1 Report
The paper aims to investigate the utility of drone wing morphometry to study spatial structure within a subspecies (here: Ibearian honeybees).
The questions/rationale are well argued in the introduction and relevant for a broader audience. The study seems to have been conducted with a good sample size and suitable approaches. However, the presentation and discussion of the results needs substantial improvements. Its currently quite difficult to understand what the results mean exactly, what the context is, what an expectation would have been for laternative scenarios. In addition, some Figures need to be fixed for basic! aspects (Fig.4/5, captions, axis entent and axis labels). A clear presentation of a spatial analysis of wing morphometry only and a visualiztion similar to Fig. 4 (to compare with the sPCA derived from SNPs or morphometry+SNPs) would benefit the paper, as one of the stated aims is to test if morphometry can be used for spatial structure analyses and hence analyses of more complex evolutionary questions.
The discussion is for a large part similar to the introduction and lacks an actual discussion of the results.
small/specific comments:
L131 "For most of the 711 colonies," specify
L133 "It was not possible to obtain five undamaged wings from drones and workers for 133 37 and 19 colonies, respectively." unclear why. so how many indiv. did you get from these colonies? four? three? one?
L137 "Wings with missing landmarks were removed" elaborate. why would there be a missing landmark?
L138: from how many of the sampled colonies?
Fig4 and parts of the main text describing the findings need to be much more explicit to clarify what is shown and make it easier to understand. Its currently unclear where the MonteCarlo simulations come from exactly and how they were generated. The red circles and the dot represent, i guess, your p-values stated in the text? But then b doesnt match the number in the text. The missing part of the axes (this should be properly shown in the graphs), make it even harder to figure out Fig4. What is the x-axis' unit?
L188 ff.
L188 - specify what global score means/ of what?,
L189 - what do these Moran I's mean - can you give an expectation or comparison with an easy to understand example. Its hard to judge what these numbers mean if completely unfamiliar with Morans I.
L191: cite the study you cie in the fig caption here.
Fig 5. Also this needs more explaination, especially in the caption. What do the numbers in the legend mean exactly?
L196, specify again what the Procrustes distances mean here (what are they supposed to show exactly?) Is this related to the previosu paragraph? Its unclear to me what the numbers mean you provide. r value around 0.5-0.6 seem to indicate not a very strong correlation. The inclusion of spatial structure seems to provide a much higher correlation. Is it possible to provide an analyses of the spatial correlation of only wingshapes, similar to the sPCA which includes the SNPs. Given the highly noisy results from Fig. 3 (giving the impression of little structuring, at least between the transects), it would be nice to actually see the effect of geography in wingshape measures without SNPs. This could be a fourth graph within Fig 4. At the moment its still unclear to me, how reliable/feasable wingshapes in a spatial sample are, when one has no SNPs to correlate. After all, this is the main focus of the paper!
The discussion lacks an actual discussion of the results of the paper. Especially at the beginning its actually more of (repetition of) an introduction.
Author Response
Reviewer 1
Comments and Suggestions for Authors
The paper aims to investigate the utility of drone wing morphometry to study spatial structure within a subspecies (here: Ibearian honeybees).
The questions/rationale are well argued in the introduction and relevant for a broader audience. The study seems to have been conducted with a good sample size and suitable approaches. However, the presentation and discussion of the results needs substantial improvements. Its currently quite difficult to understand what the results mean exactly, what the context is, what an expectation would have been for laternative scenarios. In addition, some Figures need to be fixed for basic! aspects (Fig.4/5, captions, axis entent and axis labels). A clear presentation of a spatial analysis of wing morphometry only and a visualiztion similar to Fig. 4 (to compare with the sPCA derived from SNPs or morphometry+SNPs) would benefit the paper, as one of the stated aims is to test if morphometry can be used for spatial structure analyses and hence analyses of more complex evolutionary questions.
The discussion is for a large part similar to the introduction and lacks an actual discussion of the results.
>>> Thank you for your comments and valuable suggestions. We made profound changes in the methods, results and discussion sections, in order to make a clearer presentation of our analytical approaches, our findings and their interpretation. We believe that these sections improved substantially after incorporating yours and the other reviewer suggestions. Figure 4 and the caption of figure 5 were greatly improved, as requested. The sPCA was performed using either wing shape data (Figure 4a for drones, 4b for workers) or SNPs (Figure 4c). Figure 4 is central in this paper as it shows clearly that the spatial structures inferred from the 3 datasets (wing shapes for drones in 4a, wing shapes for workers in 4b, and SNPs in 4 c) are very similar allowing a positive answer to the question “is the wing geometric morphometrics approach able to detect genetic structure in the Iberian honey bee?”. Figure 5 provides the statistical support for the structure patterns displayed in Figure 4. We never combined SNPs with morphometry, as you mentioned. We hope that this version is clearer so that the readers do not misinterpret our findings.
small/specific comments:
L131 "For most of the 711 colonies," specify
>>>It was specified in lines 147-148 in the sentence “Specifically, for 56 colonies (37 drones and 19 workers) the number of detached forewings was lower than five (see Table S1 for details).”. For clarification, we added a new table with the detailed data (please see Table S2).
L133 "It was not possible to obtain five undamaged wings from drones and workers for 133 37 and 19 colonies, respectively." unclear why. so how many indiv. did you get from these colonies? four? three? one?
>>Done. Please see the new table S2 where the detailed numbers for each colony are provided.
L137 "Wings with missing landmarks were removed" elaborate. why would there be a missing landmark?
>>Several wings lack some of the veins junctions, meaning that from the 19 landmarks one or more are missing possibly due to mutations which are more frequently expressed in the haploid drones than in the diploid workers (we addressed this issue in the discussion section). Missing landmarks is an unfortunate, common, and unavoidable problem in geometric morphometrics, which is present not only in wings but also in other morphological structures such as scallop shells, mandibles, cranial bones etc. Typically, this problem is addressed either by estimating the missing landmarks, by reducing the number of landmarks to exclude those missing from numerous specimens, or more frequently by excluding incomplete structures from analyses, as we did. The issue of missing landmarks is addressed in several works, such as:
Jessica H. Arbour and Caleb M. Brown. (2014). Incomplete specimens in geometric morphometric analyses. Methods in Ecology and Evolution 2014, 5, 16–26. Adams, D. C., Rohlf, F. J., & Slice, D. E. (2013). A field comes of age: geometric morphometrics in the 21st century.Hystrix,24(1), 7.” Mitteroecker, P., & Gunz, P. (2009). Advances in geometric morphometrics.Evolutionary Biology,36(2), 235-247.” Strauss, R.E. & Atanassov, M.N. (2006) Determining best complete subsets of specimens and characters for multivariate morphometric studies in the presence of large amounts of missing data. Biological Journal of the Linnean Society, 88, 309–328.
>>>Because missing landmarks is a common problem and removal of incomplete morphological structures is the most popular approach, we feel that our paper is not the right place to elaborate on this issue. This issue has been discussed in several methodological review articles (some listed above).
L138: from how many of the sampled colonies?
>>>Done. This information is in the new table S2 and was also added to the manuscript (please see lines 151-153).
Fig4 and parts of the main text describing the findings need to be much more explicit to clarify what is shown and make it easier to understand. Its currently unclear where the MonteCarlo simulations come from exactly and how they were generated. The red circles and the dot represent, i guess, your p-values stated in the text? But then b doesnt match the number in the text. The missing part of the axes (this should be properly shown in the graphs), make it even harder to figure out Fig4. What is the x-axis' unit?
>>>Figure 4 was renumbered to Figure 5. A better explanation of Monte Carlo simulations is provided in Material and Methods section (please see lines l84-192 ). The red circles and the black dot represent the observed max(t); All the values through the text were verified and now they are concordant and easily located in Figure 5. The caption of Figure 5 was improved to facilitate its interpretation. We improved the graphical aspects of Figure 5 to show the complete axes and their meaning. We thank you for your comments/suggestions, which contributed to a substantial improvement of the manuscript.
L188 ff.
L188 - specify what global score means/ of what?,
>>>The global score is the axis with the most positive eigenvalue. This information was added in line 212.
L189 - what do these Moran I's mean - can you give an expectation or comparison with an easy to understand example. Its hard to judge what these numbers mean if completely unfamiliar with Morans I.
>>> Moran’s I ranges from -1 to 1. Highly positive Moran’s I values indicate the presence of strong global structure, which occurs when each sampling location is genetically closer to neighbours than randomly chosen locations, as it happens in the presence of patches, clines or intermediate states. On the other hand, highly negative Moran’s I values indicate the existence of a stronger genetic differentiation among neighbours than randomly chosen locations, which indicates local structure. We added this explanation to the Material and Methods section. Please see the lines 175-182.
L191: cite the study you cie in the fig caption here.
>>>Done.
Fig 5. Also this needs more explaination, especially in the caption. What do the numbers in the legend mean exactly?
>>>Figure 5 was renumbered to figure 4 and the caption was substantially improved by including the explanation. Squares represent population scores (first principal component) and are spatially arranged according to geographical coordinates of the colonies. Large black squares indicate samples well differentiated from those denoted by large white squares, small squares represent less differentiated samples.
L196, specify again what the Procrustes distances mean here (what are they supposed to show exactly?) Is this related to the previosu paragraph? Its unclear to me what the numbers mean you provide.
>>> Pairwise Procrustes distances (Table S2) provide a measure of genetic differentiation between populations (sites) using wing shapes, after removing variations in size, position and orientation. This information was added to the first paragraph of the results section and the correlations calculated from Procrustes distances were moved from the last paragraph (in the previous version of the manuscript) to the first paragraph of the results section (in this new version). We hope that the results section is clearer now.
r value around 0.5-0.6 seem to indicate not a very strong correlation.
>>>We did not judge the correlation strength among the Procrustes distance and SNPs, we just mentioned that it is significant, as shown by a P-value<2.2X10-16. We added the word “moderate” before “significantly” (please see line 204).
The inclusion of spatial structure seems to provide a much higher correlation. Is it possible to provide an analyses of the spatial correlation of only wingshapes, similar to the sPCA which includes the SNPs.
>>>It seems that you misunderstood what we did because the analyses that you are suggesting are exactly what we did, as described in subsection “2.3. Estimation of spatial structure”. The spatial analyses were performed using only SNP data (newly renamed Figure 4c) or only morphometric data (Figure 4a for drones, Figure 4b for workers); We never used in the same analysis morphometry and SNP data together. Then we calculated correlations between the 3 different data sets (first spatial score for drones, figure 4a, first spatial score for workers, figure 4b, and first spatial score for SNPs, figure 4c). Since we changed both the methods and results sections considerably, we hope this issue is clearer now.
Given the highly noisy results from Fig. 3 (giving the impression of little structuring, at least between the transects), it would be nice to actually see the effect of geography in wingshape measures without SNPs. This could be a fourth graph within Fig 4. At the moment its still unclear to me, how reliable/feasable wingshapes in a spatial sample are, when one has no SNPs to correlate. After all, this is the main focus of the paper!
>>>This comment is related with the previous one and results from with your misunderstanding. Figure 4 shows the spatial pattern for each of the three datasets, as explained in the previous comment. As you can see in Figure 4 [and then confirm with the high correlations values computed from the first component of the sPCA (r=0.90 between SNPs (figure 4c) and drone wings (figure 4a); r=0.87 between SNPs (figure 4c) and worker wings (figure 4b)], spatial analysis of wing shapes provides a reliable representation of the genetic structure in the Iberian honey bee.
The discussion lacks an actual discussion of the results of the paper. Especially at the beginning its actually more of (repetition of) an introduction.
>>>We changed the discussion to remove overlap with the introduction. We also tried to improve the discussion by increasing our focus in the interpretation of our results. We believe that the discussion section, as well as the other sections, is much better now. We thank you for your valuable comments, which certainly contributed for a considerably improved manuscript.

Reviewer 2 Report
Reviewer comments
The Manuscript very interesting and should be published. There are some issues in discussion part and questions that should be disclosed in the text. After minor revision, improving the text and structure the Manuscript can be published.
The Manuscript has title "Genetic structure of Iberian honey bees (Apis mellifera iberiensis) inferred from workers and drones: congruence between wing geometric morphometrics and single nucleotide polymorphisms (SNPs)". According this title the discussion should be changed. Discussion part contains a lot of literature review materials about subspecies A.m.iberiensis, about advantages of wing geometric morphometrics. Authors should move all this literature review materials into introduction part. The discussion part contains scarce discussion about revealed data about Genetic structure of Iberian honey bees in current study.
Authors said that the higher similarity between SNP data and morphometry data of drones is due to SNP data revealed from drones. No SNP data revealed from workers. Authors should discuss how SNP data based on drones and workers can be related each other. It is known, that workers population totally different than drones population. Drones population may lack some alleles, which presented in workers population.
Authors should explain, why previous data based on allozymes and morphometry and mtDNA found the a gradual transition from one subspecies to another and grouped together North African with A. m. iberiensis and A. m. mellifera, whereas in SNP and wing geometric morphometry based studies African and European subspecies located separately. Are they real different and the gradual transition is result of genetic introgression between neighbour subspecies? Please explain why previous grouping was together and current grouping was separate.
Authors used the Iberian honey bee as a model system and the first their goal was to assess the efficiency of geometric morphometrics in study population substructure. Thus the title should be little changed according this goal. For example, it can be as following "The advantages of wing geometry morphometry in comparison with single nucleotide polymorphism (SNP) in study of honey bee population structure inferred from workers and drones of Iberian honey bees (Apis mellifera iberiensis)"
Please add more explanation in the text of discussion on the Figure 4. Monte Carlo simulations using a spatially explicit approach. Authors did not explain spatially explicit approach in Materials and methods. How you supported the existence of global structure in both drones (Figure 4a) and workers (4b), and no statistical evidence of local structure for either drones (Figure 4c) and workers (Figure 4d)?
In the abstract authors said "Thus, this method offers a low-cost alternative for preliminary estimation of population structure." The main goal in science is accuracy. Thus authors should say about accuracy in the first of all and only then about low-cost. The Abstract is not well, should be improved.
Conclusions of the Manuscript do not appropriate to the title and goal and should be improved.
Author Response
Reviewer 2
Comments and Suggestions for Authors
Reviewer comments
The Manuscript very interesting and should be published. There are some issues in discussion part and questions that should be disclosed in the text. After minor revision, improving the text and structure the Manuscript can be published.
>>> We deeply appreciate your comments and suggestions.
The Manuscript has title "Genetic structure of Iberian honey bees (Apis mellifera iberiensis) inferred from workers and drones: congruence between wing geometric morphometrics and single nucleotide polymorphisms (SNPs)". According this title the discussion should be changed. Discussion part contains a lot of literature review materials about subspecies A.m.iberiensis, about advantages of wing geometric morphometrics. Authors should move all this literature review materials into introduction part. The discussion part contains scarce discussion about revealed data about Genetic structure of Iberian honey bees in current study.
>>>We made profound changes across all sections, including discussion and conclusions, of the manuscript to accommodate your suggestions and those of the other reviewer. We moved most of the A. m. iberiensis literature to the introduction and we focused more in discussing our findings, as suggested. We also changed the title of the manuscript to better reflect the focus of the study.
Authors said that the higher similarity between SNP data and morphometry data of drones is due to SNP data revealed from drones. No SNP data revealed from workers. Authors should discuss how SNP data based on drones and workers can be related each other. It is known, that workers population totally different than drones population. Drones population may lack some alleles, which presented in workers population.
>>>Unfortunately, due to budget constraints, we only have SNP data for drones of the 711 colonies. [We published the main results of the SNPs in two publications in Molecular Ecology (Chavéz-Galarza et al. 2013, 2015)]. Therefore, we can’t discuss how SNP data from workers would relate to SNP data from drones. Besides, the emphasis of this paper is on how the structure retrieved by morphometry relates to the structure retrieved by molecular data and not on how the structure retrieved by SNPs of drones relate to that retrieved by SNPs of workers. In any case, based upon our experience with SNPs (we have published already several papers using SNPs in A.m. iberiensis and in other subspecies), and from what we have learned from the literature and from our previous works on the Iberian honey bee, we are certain that the patterns retrieved from SNPs genotyped in workers would be similar to those retrieved from our drone SNPs. In our study published in Molecular Ecology (Chávez‐Galarza et al. Revisiting the Iberian honey bee (Apis mellifera iberiensis) contact zone: maternal and genome‐wide nuclear variations provide support for secondary contact from historical refugia. Molecular Ecology 2015, 24, 2973-2992.), we compared the genetic patterns retrieved from the drone SNPs with those from the mtDNA for the same 711 colonies. Interestingly, even though the mtDNA is maternally inherited and the SNPs are bi-parentally inherited, we found highly correlated patterns between these two different markers (see Figure 3 in Chávez-Galarza et al. 2015). So, having SNP data from workers wouldn’t add much to one of the main questions of this paper: “Is the wing geometric morphometrics approach able to detect genetic structure in the Iberian honey bee? “. The answer is clearly yes and would not change if the comparisons were done with SNPs from workers.
>>>Typically, SNPs are biallelic, which means that if a queen is heterozygous let’s say for AG then her female offspring will receive either A or G and the drone offspring will also receive either A or G. The difference between the workers and the drones is that the workers will receive another allele from their fathers (which will be A or G because of the biallelic nature of the SNPs and depending on the frequency of A and G in the population) whereas the drones won’t receive another allele because they are haploid.
>>>You mention that “It is known, that workers population totally different than drones population. Drones population may lack some alleles, which presented in workers population.” We are sorry, but we do not know these works and we are very puzzled about why the frequency of autosomal loci would be different between sexes. We would be happy to see those works to better understand your point.
Authors should explain, why previous data based on allozymes and morphometry and mtDNA found the a gradual transition from one subspecies to another and grouped together North African with A. m. iberiensis and A. m. mellifera, whereas in SNP and wing geometric morphometry based studies African and European subspecies located separately. Are they real different and the gradual transition is result of genetic introgression between neighbour subspecies? Please explain why previous grouping was together and current grouping was separate.
>>>It is common knowledge in the field of population genetics that different markers may provide different results, depending on whether they are neutral or under selection, depending on whether they are bi-parentally or uni-parentally (mtDNA) inherited, and in this case the effective size is 4 times lower and therefore mtDNA loci are more prone to genetic drift, depending on gene flow is male-directed or female-directed or not etc. These issues have been repeatedly addressed in many works, but see for example: “Freeland, J., H. Kirk, and S. Petersen. Molecular markers in ecology. Molecular Ecology.(Ed. H. Kirk.) pp (2005): 31-62, “Sunnucks, Paul. Efficient genetic markers for population biology. Trends in ecology & evolution 15.5 (2000): 199-203”.
>>> The issue of why allozymes and morphometry provide different results from microsatellites and SNPs and mtDNA in honey bees is out of the scope of this study. Our aim was to verify if geometric morphometric data provide genetic patterns similar to those of SNPs. Addressing this issue in this manuscript would divert the reader from the main focus of the paper and would make the introduction even longer. Besides, this issue has been repeatedly addressed and discussed in our papers on the Iberian honey bee [see for example “Chávez‐Galarza, J., Henriques, D., Johnston, J. S., Carneiro, M., Rufino, J., Patton, J. C., & Pinto, M. A. (2015). Revisiting the Iberian honey bee (Apis mellifera iberiensis) contact zone: maternal and genome‐wide nuclear variations provide support for secondary contact from historical refugia. Molecular ecology, 24(12), 2973-2992” and Chávez‐Galarza, J.; Henriques, D.; Johnston, J.S.; Azevedo, J.C.; Patton, J.C.; Muñoz, I.; De la Rúa, P.; Pinto, M.A. Signatures of selection in the Iberian honey bee (Apis mellifera iberiensis) revealed by a genome scan analysis of single nucleotide polymorphisms. Molecular Ecology 2013, 22, 5890-5907] and in evolutionary honey bee papers published by the French group led by Lionel Garnery (some of their papers are listed in the references).
Authors used the Iberian honey bee as a model system and the first their goal was to assess the efficiency of geometric morphometrics in study population substructure. Thus the title should be little changed according this goal. For example, it can be as following "The advantages of wing geometry morphometry in comparison with single nucleotide polymorphism (SNP) in study of honey bee population structure inferred from workers and drones of Iberian honey bees (Apis mellifera iberiensis)"
>>> We changed the title to better reflect the two goals of the paper and the main findings. The title is now:” Wing geometric morphometrics of workers and drones and single nucleotide polymorphisms provide similar genetic structure in the Iberian honey bee (Apis mellifera iberiensis)”.
Please add more explanation in the text of discussion on the Figure 4. Monte Carlo simulations using a spatially explicit approach. Authors did not explain spatially explicit approach in Materials and methods. How you supported the existence of global structure in both drones (Figure 4a) and workers (4b), and no statistical evidence of local structure for either drones (Figure 4c) and workers (Figure 4d)?
>>>We changed the discussion to better reflect the results obtained from the spatial analysis, which detected significant (P-value is generated by the Monte Carlo simulations) global structure (please see line 265-269). We added more information about sPCA and Monte Carlo simulations in Material and Methods (please see lines 174-192) and changed dramatically the results section (including the captions of newly renamed figures 4 and 5) for a clearer presentation of the sPCA findings. There is support for global structure in drones and workers because the observed max(t) associated to global structure is outside of the simulated distribution and corresponds to a P-value ≤1x10-4, meaning that the spatial distribution of the genetic variance is not random and the null hypothesis can be rejected. On the other hand, there is no statistical evidence of local structure for either drones and workers because the observed max(t) associated with local structure is inside the simulated distribution with a P-value ≥1x104.
In the abstract authors said "Thus, this method offers a low-cost alternative for preliminary estimation of population structure." The main goal in science is accuracy. Thus authors should say about accuracy in the first of all and only then about low-cost. The Abstract is not well, should be improved.
>> >We changed the last sentence to emphasize that geometric morphometrics is a reliable and low-cost alternative to molecular methods. We prefer to use the word “reliable” instead of “accurate” because accuracy has a specific meaning in statistics. Since we did not specifically assess accuracy in this paper (we did that in other SNP papers, see for example: (1) Henriques et al. 2018. High sample throughput genotyping for estimating C-lineage introgression in the dark honeybee: an accurate and cost-effective SNP-based tool. Scientific Reports 8 (1): 8552. DOI: 10.1038/s41598-018-26932-1; or (2) Henriques et al. 2018. Developing reduced SNP assays from whole-genome sequence data to estimate introgression in an organism with complex genetic patterns, the Iberian honeybee (Apis mellifera iberiensis). Evolutionary Applications 11: 1270–1282. DOI: 10.1111/eva.12623), we prefer to avoid the word “accuracy”.
>>>We made considerable changes in the abstract. We hope you like it better now. Since you didn’t specify what was wrong and should be changed, there is a chance that this new version is still far from what you expected from an abstract. It is worthwhile noting, however, that we tried to follow the guidelines provided by the Journal: “follow the style of structured abstracts, but without headings: 1) Background: Place the question addressed in a broad context and highlight the purpose of the study; 2) Methods: Describe briefly the main methods or treatments applied. Include any relevant preregistration numbers, and species and strains of any animals used. 3) Results: Summarize the article's main findings; and 4) Conclusion: Indicate the main conclusions or interpretations. The abstract should be an objective representation of the article:”
>>>We thank you for your valuable comments, which certainly contributed for a considerably improved manuscript.